# Rapid and Accurate Crayfish Sorting by Size and Maturity Based on Improved YOLOv5

**Xuhui Ye** [1,†], **Yuxiang Liu** [1,†], **Daode Zhang** [1,*], **Xinyu Hu** [1], **Zhuang He** [1] and **Yan Chen** [2]

1   School of Mechanical Engineering, Hubei University of Technology, Wuhan 430068, China; yxh89@hbut.edu.cn (X.Y.); 102110154@hbut.edu.cn (Y.L.); 19991012@hbut.edu.cn (X.H.); 102110170@hbut.edu.cn (Z.H.)
2   School of Mechanical and Electric Engineering, Wuhan Donghu University, Wuhan 430212, China; chenyandhxy@139.com
*   Correspondence: 19951044@hbut.edu.cn
†   These authors contributed equally to this work.

**Abstract:** In response to the issues of high-intensity labor, low efficiency, and potential damage to crayfish associated with traditional manual sorting methods, an automated and non-contact sorting approach based on an improved YOLOv5 algorithm is proposed for the rapid sorting of crayfish maturity and size. To address the difficulty in focusing on small crayfish, the Backbone is augmented with Coordinate Attention to boost its capability to extract features. Additionally, to address the difficulty in achieving high overall algorithm efficiency and reducing feature redundancy, the Bottleneck Transformer is integrated into both the Backbone and Neck, which improves the accuracy, generalization performance, and the model's computational proficiency. The dataset of 3464 images of crayfish collected from a crayfish breeding farm is used for the experiments. The dataset is partitioned randomly, with 80% of the data used for training and the remaining 20% used for testing. The results indicate that the proposed algorithm achieves an mAP of 98.8%. Finally, the model is deployed using TensorRT, and the processing time for an image is reduced to just 2 ms, which greatly improves the processing speed of the model. In conclusion, this approach provides an accurate, efficient, fast, and automated solution for crayfish sorting.

**Keywords:** crayfish sorting; Yolov5; attention mechanism; maturity; size; TensorRT





## 1. Introduction

The crayfish is a freshwater crustacean mainly found in southern China and the southern United States. Given its unique sensory, textural, and flavor characteristics, cray-fish has emerged as a highly coveted and prized delicacy in the culinary world. As a longstanding culinary tradition in China, crayfish has also gained popularity and recognition in international gastronomy. With the continuous expansion of the crayfish market, this species has become a significant representative of Chinese food culture and has stimulated the growth of crayfish farming as a major industry. In 2021, crayfish farming in China encompassed a total area of 26 million mu, yielding a production volume of 2.6336 million tons, thereby solidifying its position as the sixth largest freshwater aquatic product in the country, ranking only behind major freshwater fish [1].

Conventional crayfish classification methods have typically relied on manual sorting, which is associated with high labor costs and significant resource requirements. Moreover, these methods often result in damage to the crayfish, which further limits their utility for businesses that prioritize production efficiency and product quality. As a result, automated and non-contact sorting techniques have become an indispensable component of the crayfish production process. Over the past few years, computer vision has made significant strides [2–4]. As an advanced technology, computer vision not only enhances production efficiency and product quality, but also reduces labor costs [5]. In contrast to conventional

methods, computer vision technology offers unparalleled advantages in measuring non-linear relationships, perimeters, areas, quantities, colors, and chemical compositions of the target object [6–9]. Ma [10] utilized the K-means clustering algorithm to segment color images containing groupers. Through segmentation experiments on 100 artificial seawater images of groupers in RGB format, they achieved a high accuracy of 98 blue components. However, this approach is only feasible in underwater settings and not well suited for intricate backgrounds. Kesvarakul [11] put forward a method for shrimp larva counting using image detection for spot information, which can reduce errors by 6.9% compared to traditional manual counting. However, this method requires comparison of the transparent parts of the shrimp larvae with the surrounding water environment, and the accuracy of counting may be interfered with by water impurities and changes in appearance, and parameters need to be adjusted as the aquaculture environment changes. JM [12] proposed an embedded system that is easy to operate, low cost, and has a high precision for counting fish schools, which utilizes the connected component relationship of the fish body area and perimeter in images. However, this method may be affected by mutual occlusion when fish density is high. Zhu [13] applied a stacked denoising autoencoder (SdA) to extract effective features from the sensor responses of a machine olfaction system. They further performed qualitative classification using a support vector machine (SVM). The study analyzed the total volatile basic nitrogen and total viable count of crabs over the course of their preservation. The experimental results demonstrate that the recognition rate can reach 96.67%.

As mentioned above, traditional machine learning algorithms have become difficult to meet the real-time requirements. The past few years have witnessed a surge in the use of deep learning technology in aquatic animal research [14,15]. On the one hand, the powerful neural network structure of deep learning can process large-scale animal images and data [16], and extract effective features from them. On the other hand, with the continuous improvement of computer performance and hardware devices, deep learning technology has become more efficient, accurate, and has strong generalization ability [17]. These fac-tors have made deep learning an important tool in the fields of aquaculture and aquatic product detection. Li [18] used binocular stereo vision technology to obtain three-dimensional information and performed fish detection and fine segmentation using the Mask-RCNN network. Finally, three-dimensional point cloud data of the fish surface were generated, and the external dimensions of multiple fish under free movement were calculated. Sun [19] addressed the issue of limited generalization ability in existing multi-object fish detection methods that mostly focus on controlled environments. They proposed a transfer learning approach based on DRN for feature extraction from raw images, combined with RPN to generate candidate detection boxes. Using Faster-RCNN, they created a fish detection model capable of detecting multiple objects in complex back-grounds. By testing in complex backgrounds, the model was found to achieve an average detection accuracy of 89.5% for goldfish. To tackle challenges posed by blurriness and the multi-degree-of-freedom motion of objects in water, Xu [20] proposed a YOLO-V3 algorithm-based model for object recognition. The model successfully tracked objects with a multi-degree-of-freedom motion in water, achieving an average accuracy of 75.1% when the confidence level was set at 0.5. Wageeh [21] proposed a cost-effective and easy-to-operate monitoring method based on YOLO for effectively detecting and counting fish in water using image enhancement and object detection algorithms, and extracting trajectory features to enhance the tracking and detection capabilities of fish ponds in aquaculture. For the specific features of sea urchin spines, Hu [22] developed a feature-enhanced sea urchin detection algorithm that used a multi-directional edge detection algorithm for feature enhancement, utilized ResNet 50 as the basic framework, and employed a feature-level fusion technique to improve feature extraction capability and semantic representation. Experimental results showed that the proposed algorithm achieved a 7.6% improvement in AP value over the SSD algorithm and improved the confidence score for small targets. To address the problems of underwater image blur and difficulty in capturing small targets,

Hu [23] proposed an approach that utilizes an optimized YOLO-V4 to detect uneaten feed particles in aquaculture. The proposed method enhanced the network performance and model accuracy by modifying the way the feature pyramid network and path aggregation network are connected, as well as the residual connection mode. The experiments showed an average precision improvement of 27.21%. To track crayfish from capture to consumers, Vo [24] proposed an image-based individual recognition solution using convolutional neural networks. This approach employed a combination of the Siamese model and a contrastive loss function to distinguish individual crayfish by analyzing their exoskeleton images, providing a more secure and reliable approach for crayfish recognition and tracking.

The convolutional neural network-based methods mentioned above usually incorporate attention mechanisms or increase the network depth to enhance detection accuracy. However, the model's parameters increase with this approach, leading to slower processing speeds that fail to satisfy the requirement for detecting in real time. This method faces challenges such as difficulty in live crayfish sorting, insufficient accuracy, and low efficiency. Therefore, a balance between improving accuracy and maintaining high-speed processing is necessary. The proposed method, which utilizes an improved YOLOv5, was evaluated on a crayfish dataset through experiments, for the fast sorting of crayfish maturity and size.

Therefore, to achieve automated, contactless, fast, and accurate sorting of live crayfish, we propose the following methods to address this objective:

1.  Achieve lightweight and improved accuracy and generalization performance by introducing the Bottleneck transformer structure and decreasing the number of Backbone modules.
2.  The Bottleneck transformer structure is introduced in the Neck's head to better address the scale problem of feature maps, fuse more feature information, and address the problem of small inter-class differences.
3.  The incorporation of the Coordinate Attention module into the Backbone enhances both the channel and position awareness of the network, resulting in improved extraction of features.
4.  To enhance the improved model, we deploy it using TensorRT, enabling extremely fast inference and prediction sorting while ensuring accuracy.

## 2. Materials

In this study, the research was approximately 10 kg of crayfish purchased from a crayfish farm. The images were captured on a conveyor belt at the crayfish farm, as shown in Figure 1, using a Hikvision industrial camera (MV-CS016-10GC) and a Hikvision robot FA lens (MVL-MF0828M-8MP). The camera was fixed in a position perpendicular to the conveyor belt and directly above the crayfish, with a distance of 51 cm and a field of view of 23 cm by 31.68 cm. Two LED strip lights (BRD24030) were placed above the conveyor belt on both sides of the crayfish for stable illumination during image capture.

Crayfish can be divided into green and red based on their maturity. The crayfish shells are composed of several layers of different tissues, including the outer horny layer, the middle pigment layer, and the inner transparent membrane-like layer. These layers not only provide protection and support for the crayfish but also reflect or absorb light of different wavelengths, resulting in different colors. Red crayfish typically contain a higher concentration of red pigments in the pigment layer, which can absorb blue and green light and reflect red and yellow light, resulting in a vivid red color. Therefore, when light passes through the red crayfish's shells and goes through the pigment layer, red and yellow light will be reflected while blue and green light will be absorbed, making the red crayfish's shells typically opaque. The shells of green crayfish are usually not completely transparent but contain tiny crystal structures in the membrane-like layer that can scatter and reflect light. These small structures can make light appear in different colors, including blue and green, so the shells of green crayfish usually appear deep green or blue–green. When light passes through the green crayfish's shells and goes through these tiny crystal

structures, the light is scattered and reflected, making the shells of green crayfish typically more transparent than the shells of red crayfish.

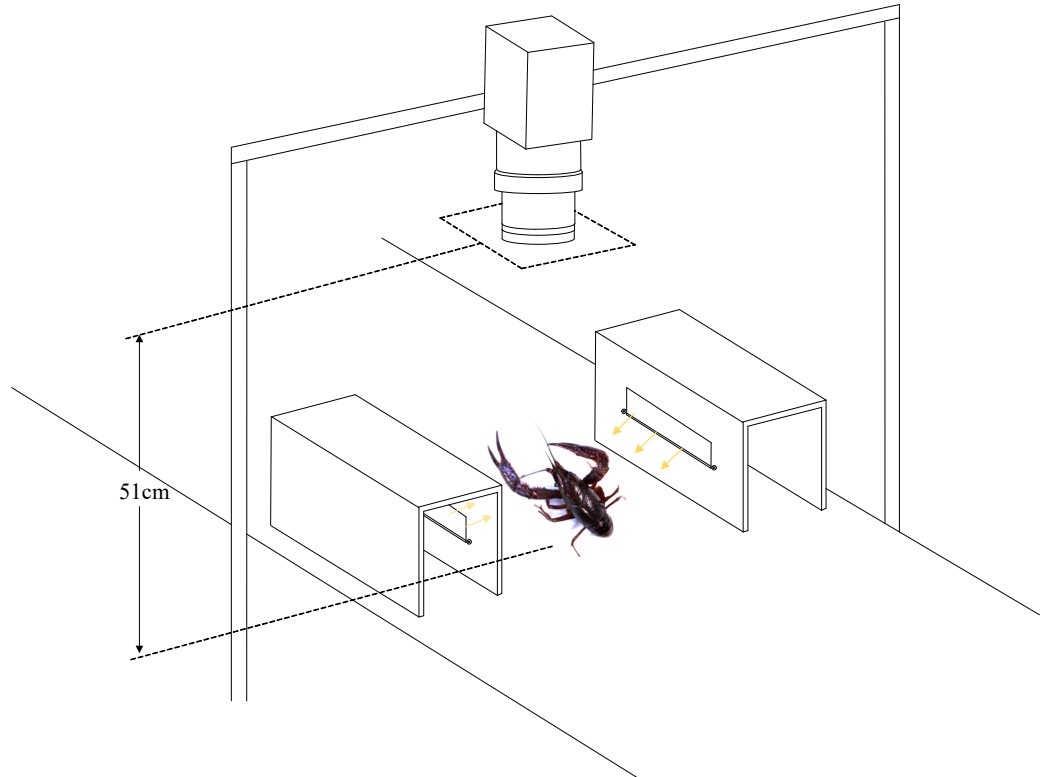

**Figure 1.** Graphical representation of the process for obtaining images of crayfish.

Table 1 presents the relevant information regarding the crayfish dataset. The sizes are divided into four levels: small (<20 g), medium (20~30 g), large (30~40 g), and extra large (>40 g). In order to facilitate the labeling process, the actual labels are 01, 02, 03, and 04, where 01, 02, 03, and 04 correspond to small, medium, large, and extra large, respectively. In this study, a total of 3464 test images were obtained, with a resolution of 720 pixels (horizontal) × 540 pixels (vertical), and were classified into 8 categories, including 477 images classified as red_small, 450 images classified as red_medium, 456 images classified as red_large, 265 images classified as red_extralarge, 721 images classified as green_small, 496 images classified as green_medium, 457 images classified as green_large, and 142 images classified as green_extralarge. The crayfish were manually labeled using LabelImg to generate xml and txt files. Figure 2 shows the annotation results of the crayfish dataset. Among the 3464 test images, 2725 images (80% of the dataset) were used as the training set, and 734 images (20% of the dataset) were used as the testing set to test and validate the detection method.

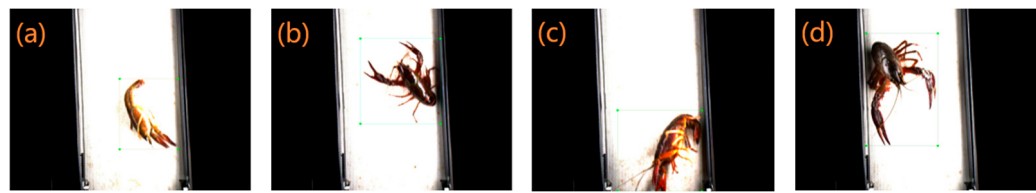

**Figure 2.** *Cont.*

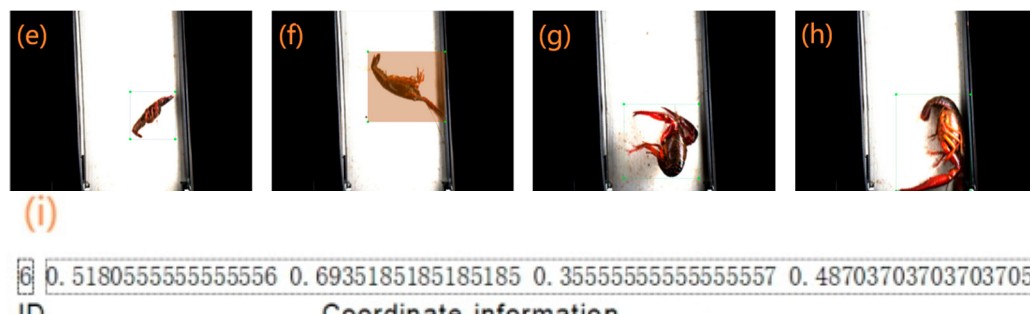

**Figure 2.** (**a**–**h**) Images represent the annotation results for green, small crayfish; green, medium crayfish; green, large crayfish; green, extra-large crayfish; red, small crayfish; red, medium crayfish; red, large crayfish; and red, extra-large crayfish; (**i**) represents the ID and coordinate information of the annotation of red, large crayfish in (**c**); the category corresponding to the ID is the sixth category, that is, the red, medium crayfish, coordinate information refers to the normalized coordinates of the bounding box.

**Table 1.** Crayfish dataset.

| Crayfish Dataset (3464 Images) | | | | 720 (Pixels) × 540 (Pixels) |
|---|---|---|---|---|
| Train Set (2725 images) | Red crayfish (1296 images) | | | |
| | Small (363 images) | Medium (357 images) | Large (365 images) | Extra large (211 images) |
| | Green crayfish (1433 images) | | | |
| | Small (584 images) | Medium (373 images) | Large (364 images) | Extra large (112 images) |
| Test Set (734 images) | Red crayfish (352 images) | | | |
| | Small (114 images) | Medium (93 images) | Large (91 images) | Extra large (54 images) |
| | Green crayfish (383 images) | | | |
| | Small (137 images) | Medium (123 images) | Large (93 images) | Extra large (30 images) |

## 3. Method

### 3.1. YOLOv5

Developed by Ultralytics LLC, YOLOv5 [25] is an object detection algorithm that utilizes deep learning. The lightweight network architecture of YOLOv5 is composed of three main components: feature extraction, fusion, and detection. Specifically, the feature extraction component serves as the Backbone network, the fusion component integrates multi-level features, and the detection component produces bounding boxes and class probabilities. Figure 3 depicts the structure of YOLOv5. The YOLOv5 Backbone architecture primarily consists of Convolution (Conv), Cross-Stage Partial Network (CSP) [26], and Spatial Pyramid Pooling Fast (SPPF). Within Conv, Conv2d, Batch Normalization, and Swish activation function are the primary components. As a critical feature extraction module in YOLOv5, the CSP module ensures the consistency of input and output, reduces the amount of calculation while improving the detection speed, and maintains good detection performance. The SPPF module replaces the single large pooling kernel used in the SPP module with multiple small kernels cascaded in a pyramid structure, which preserves the feature fusion capability and enhances the feature representation ability while further improving the processing speed.

The Neck part of YOLOv5's feature fusion refers to the Feature Pyramid Network (FPN) [27] and the Path Aggregation Network (PANet) [28]. The Feature Pyramid Network (FPN) proposes an effective solution, which is to propagate the semantic information from deep features to shallow ones and improve the expression of multi-scale features' semantics. PANet is a multi-scale feature fusion method. Based on FPN, PANet introduces a bottom-up pathway that enables both top-down and bottom-up feature fusion with lateral connections. This enhances the representation of position and semantic information at various output levels, thereby optimizing the model's feature extraction capability.

The Head section of YOLOv5 serves as the output layer of the object detection model, responsible for predicting the class and location of objects. For each feature map, the Head section expands the channel number and uses $1 \times 1$ convolution to expand the channel number to (number of classes + 5) × (number of anchors per detection layer), where 5 corresponds to the predicted bounding box's center coordinates, width, height, and confidence score. The final predicted boxes are obtained by applying non-maximum suppression (NMS) [29] to eliminate redundant detection boxes.

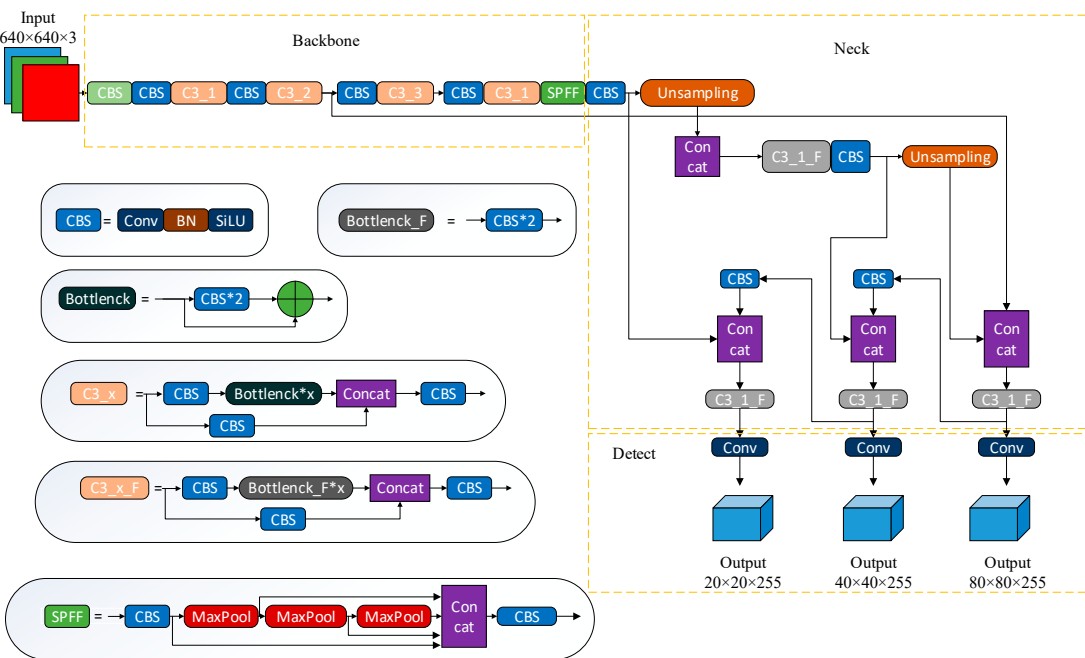

**Figure 3.** The structure of YOLOv5: Backbone, Neck, and Head components.

### 3.2. Improved YOLOv5

During the transportation of crayfish on a conveyor belt, it is difficult to pay attention to their size and maturity. Moreover, the inter-class variation [30] between crayfish is not significant. The YOLOv5 model is improved in this paper to propose a fast and accurate method to object detection. The specific method of this paper is as follows. Firstly, we reduce the number of modules in the Backbone and introduce the Bottleneck transformer structure, which not only makes the algorithm more lightweight, but also improves its accuracy and generalization performance. Secondly, we introduce the Bottleneck transformer structure into the head of the Neck to better handle the scale of feature maps, fuse more feature information, and solve the problem of small inter-class variation. Thirdly, we embed a Coordinate Attention (CA) mechanism within the Backbone structure to enhance channel and position information, and improve the feature extraction capability of Backbone. Figure 4 illustrates the improved YOLOv5 network.

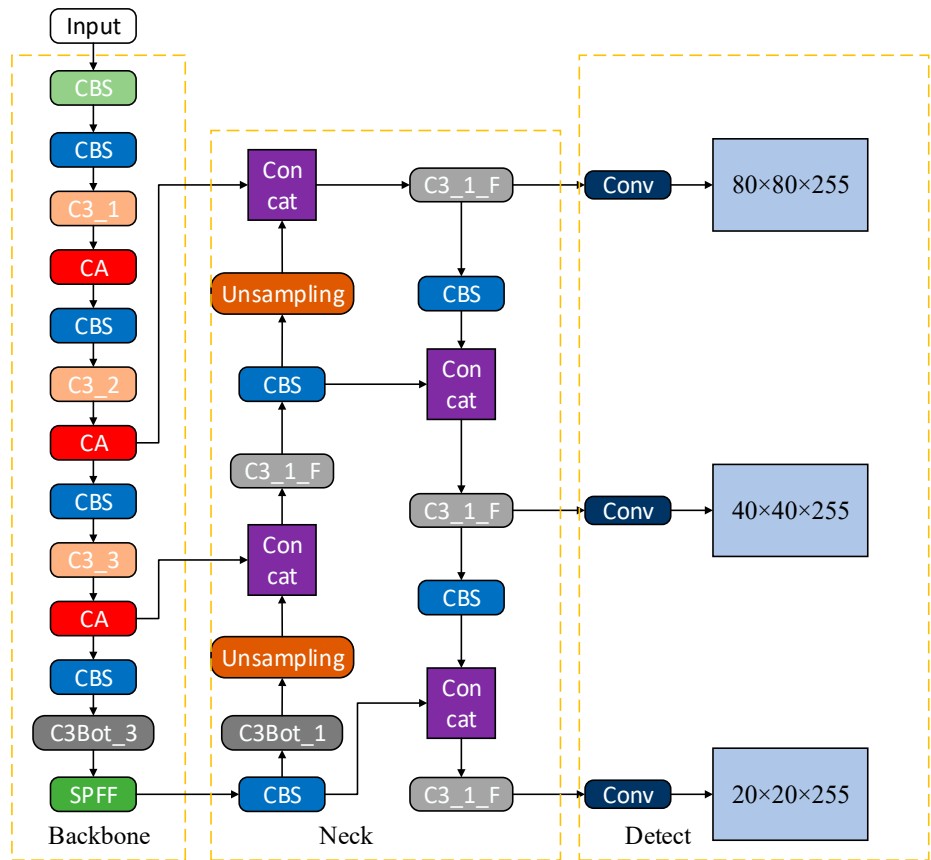

**Figure 4.** The improved YOLOv5: Lightweight Backbone, Enhanced Neck with Bottleneck Transformer, and Coordinate Attention.

### 3.2.1. Bottleneck Transformers

The Bottleneck Transformers Network (BoTNet) [31] is a hybrid model combining convolutional and self-attention mechanisms (CNN + Self-Attention). CNN has translational invariance and locality, while Transformer has a global receptive field. By integrating CNN and Transformer, the CNN + Transformer architecture leverages the strengths of both to improve object detection performance. BoTNet utilizes this advantage by replacing the 3 × 3 convolutions in the bottleneck of ResNet50 with Multi-Head Self-Attention (MHSA), as shown in Figure 5.

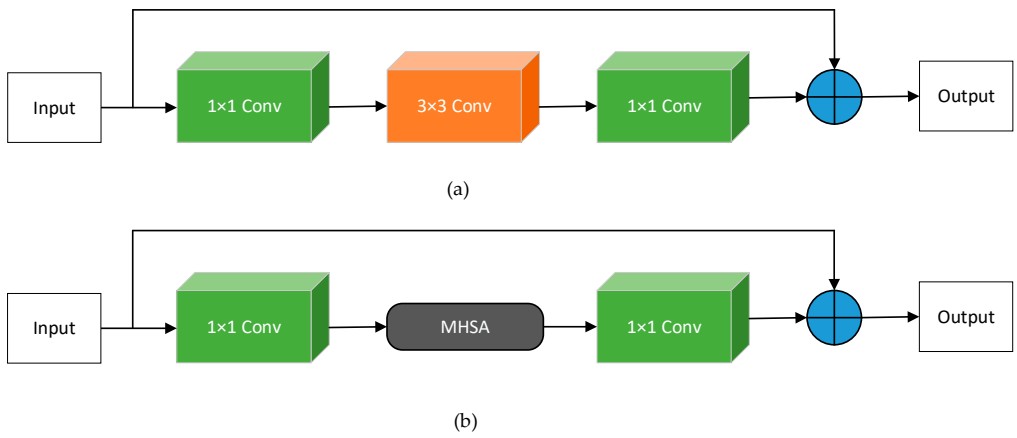

**Figure 5.** (**a**) ResNet Bottleneck; (**b**) Bottleneck Transformer.

The Multi-Head Self-Attention (MHSA) is a key attention mechanism in BoTNet, as shown in Figure 6. MHSA takes an input feature matrix of dimensions H × W × d, where H and W denote the height and width, and d represents a single token's dimension. The attention score is qkT + qrT and incorporates the query (q), key (k), and positional encoding (r) (using relative distance encoding). These vectors are broadcasted to all positions in the feature map, and then the two d-dimensional vectors corresponding to positions (i, j) are added element wise to obtain a single (H + W) × d-dimensional vector. This concatenated vector is then used for element-wise summation and matrix multiplication, followed by matrix multiplication with query matrix to generate the attention, which is finally obtained by using SoftMax over the sum of query and key vectors. The BoTNet is embedded into the Backbone of the object detection model by replacing the last C3 structure in the Backbone with the BoTNet structure, which improves the baseline and enhances the performance while keeping the model lightweight.

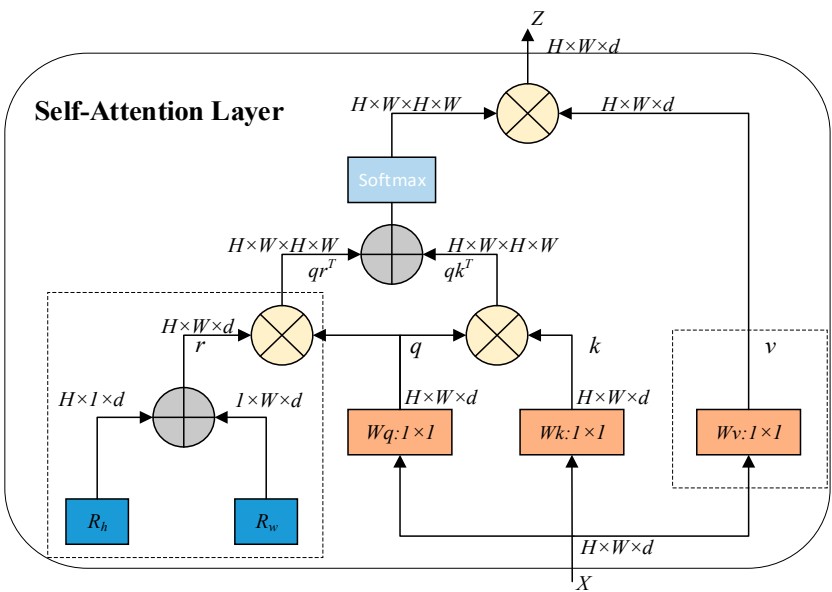

**Figure 6.** MHSA layer utilized in BoT module.

3.2.2. Coordinate Attention

Channel attention has become popular in object detection [32], as seen in SE [33] and CBAM [34]. While SE attention focuses only on the inter-dependencies between channels, neglecting the importance of spatial information, and CBAM attention considers local position information through reducing the input tensor's channel dimensionality and using convolution to obtain local information, it falls short in capturing long-range dependencies present in the feature map. Therefore, this paper introduces the Coordinate Attention (CA) [35], which optimizes the structure of the attention mechanism by incorporating position information while attending to channel information, directional sensitivity, and position information. This mechanism not only achieves fast and accurate localization of the target object but also reduces the model's complexity. Figure 7 shows the Coordinate Attention structure.

In Coordinate Attention, global average pooling is divided into two parallel 1D feature encodings, which are aggregated horizontally and vertically. This process produces two independent feature maps that incorporate positional information and capture long-range dependencies in a direction-sensitive manner. The Coordinate Attention module comprises two steps. First, embedding coordinates preserves the spatial position information while effectively encoding channel relationships and capturing long-term dependencies. Then, generating Coordinate Attention captures long-range dependencies and channel relationships by incorporating the embedded positional information. To achieve precise positional information for remote spatial interaction within the attention module, a pair of 1D feature

encodings are employed. To encode each channel's information along the horizontal and vertical coordinates, an (H, 1) or (1, W) pooling kernel is utilized on the given input $x_c$, as shown in Equation (1). The c-th channel generates an output that possesses a height of h and a width of ω, which is given by Equations (2) and (3). By aggregating features along both spatial directions and retaining spatial position information, the Coordinate Attention module achieves remote spatial interaction and improves the model's accuracy in locating objects of interest.

$$Y_c = \frac{1}{H \times W} \sum_{m=1}^{H} \sum_{n=1}^{W} x_c(m,n) \tag{1}$$

$$Y_h = \frac{1}{W} \sum_{0 \leq m < W} x_c(h,m) \tag{2}$$

$$Y_w = \frac{1}{H} \sum_{0 \leq n < H} x_c(n,w) \tag{3}$$

Combining the structure illustrated in Figure 7, in the above equation, the input $x_c$ is obtained directly from a convolutional layer, which employs a fixed kernel size. In the current attention module, h and ω signify the dimensions of the input feature map, the size of the pooling kernel is denoted by H and W. $Y_c$, $Y_h$, and $Y_w$ correspond to the outputs related to the c-th channel, where $Y_h$ represents the output corresponding to height h and $Y_w$ represents the output corresponding to width ω.

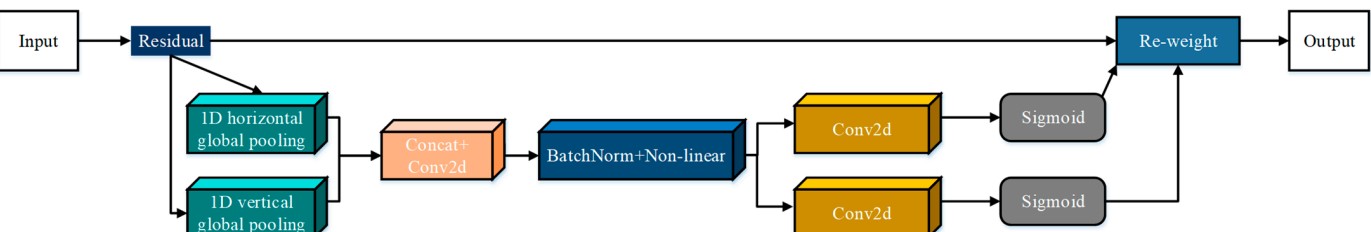

**Figure 7.** Coordinate Attention structure: capturing spatial dependencies for enhanced object localization.

After embedding the coordinate information, the two feature maps are combined through concatenation and the number of channels is reduced by applying compression with a Conv2d layer. The information regarding spatial relationships in both the horizontal and vertical orientations is encoded, and the results undergo parallel Conv2d layers to enhance the channel dimensions, followed by the application of a Sigmoid activation function to introduce nonlinearity, and the final Coordinate Attention is obtained by the multiplication of the corresponding elements between the attention maps and the input feature map.

### 3.3. Accelerating Networks with TensorRT

TensorRT is a high-performance tool focused on optimizing deep learning inference. It aims to accelerate deep learning applications on various scales of data centers, embedded platforms, and autonomous driving platforms, and supports the majority of mainstream deep learning frameworks. TensorRT can map network models from these frameworks to corresponding layers in TensorRT for efficient deployment and inference. It can also convert models from other frameworks into TensorRT models and utilize NVIDIA GPUs for optimization and acceleration. TensorRT is partitioned into two stages: compilation and deployment. The flowchart of TensorRT is shown in Figure 8. Batch input data are inferred through the plan file. Specifically, the enhanced YOLOv5 model (.pt) is first converted into a model format (.wts) using Python, facilitating efficient storage and utilization. Subsequently, the CMake command is executed with tensorrtx as the input, resulting in the generation of a serializable file compatible with Visual Studio. The generated project is then built

to produce an executable file (.exe). Additionally, the (.wts) model is converted into the (.engine) format using C language, which can be read by TensorRT. Regarding the model deployment, a detection file is created within the generated project, encompassing model construction, initialization, inference, and output processes. This optimized and deployed model showcases a high-performance integration, utilizing a serializable plan file to achieve advanced accuracy and speed.

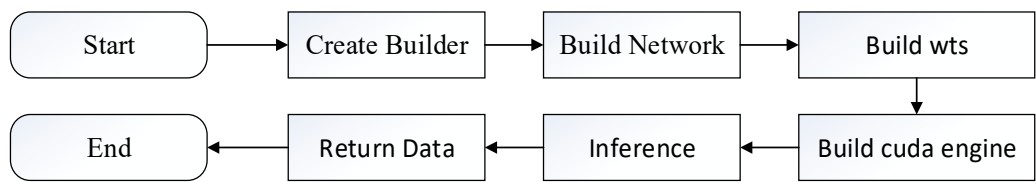

**Figure 8.** The process of TensorRT.

TensorRT applies several optimization strategies to reduce computation and memory usage. The optimization strategies of TensorRT are shown in Figure 9, where the Inception structure represents the original network. Firstly, it eliminates unused layers and operations that are equivalent to no-op. Secondly, it supports fusion of convolutional layers, biases, and ReLU operations to further reduce memory usage and data transfer. Vertical op-timization was performed compared to the Inception structure by incorporating fusion optimization of Conv + Bias + ReLU operations. Thirdly, it aggregates operations with similar parameters and target tensors to further reduce computation and memory usage. Horizontal optimization was conducted relative to vertical fusion by consolidating all 1 × 1 Conv + Bias + ReLU (CBR) operations into a single large CBR module. Finally, it can directly route the output to the correct final destination to merge concatenation layers. In comparison to horizontal fusion, the concatenation layer was directly eliminated, enabling the direct transmission of inputs from the concatenation layer to the next input. This reduction eliminated one transfer throughput and improved overall efficiency.

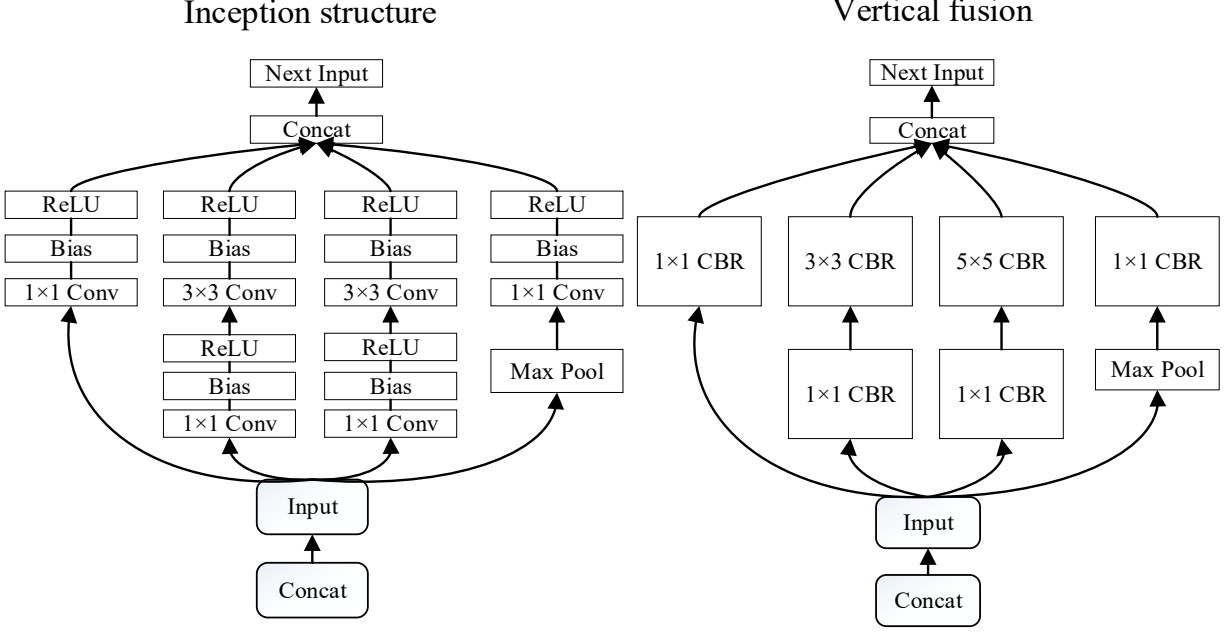

**Figure 9.** *Cont.*

Horizontal fusion

TensorRT optimized Network

**Figure 9.** Optimization strategies of TensorRT.

## 4. Results and Analysis

### 4.1. Experimental Platform

The experimental platform is based on Windows, Python 3.9, and PyTorch 1.13.1 deep learning frameworks. The hardware setup in the study consists of an octa-core Intel Core i7-10700 processor running at 2.90 GHz, 16 GB of RAM, a 240 GB solid-state drive, and a 16 GB NVIDIA RTX A4000 GPU. The experiment was developed using Microsoft Visual Studio 2017, with CuDNN version 8.3.3, CUDA version 11.7, TensorRT version 8.5.1.7, Opencv Dnn 4.5.5, Openvino 2022.1.0, and Onnxruntime-1.9.0. Table 2 presents the model parameters.

**Table 2.** Model training parameter settings.

| Parameter | Value |
|---|---|
| Batch Size | 64 |
| Image size | $640 \times 640$ |
| Epochs | 300 |
| Momentum | 0.937 |
| Optimizer | SGD |
| Initial learning rate | 0.01 |
| Final learning rate | 0.1 |

### 4.2. Evaluation Metrics

To validate the effectiveness of the presented model, we employed the following evaluation metrics: precision (P), recall (R), model size, and mean average precision (mAP) [36]. The formulas are presented below:

$$Precision = \frac{TP}{TP + FP} \tag{4}$$

$$Recall = \frac{TP}{TP + FN} \tag{5}$$

$$AP = \int_0^1 Max(\mathrm{P(k)})\mathrm{d}k \tag{6}$$

$$mAP = \frac{1}{m} \sum AP(j) \tag{7}$$

In the above equation, TP, FP, and FN signify the quantity of true positives, false positives, and false negatives, respectively. Max(p(k)) represents the maximum precision value at point k, m represents the quantity of categories in the test set, and AP(j) represents the average precision of the j-th category.

Accuracy and recall are the most basic metrics for evaluating the performance of object detection algorithms. Accuracy denotes the correct identification rate for positive samples among all detected positives, while recall indicates the accurate detection rate for positive samples among all true positives. In object detection, positive samples refer to detected targets. Model size is usually used to measure the complexity and deploy the ability of algorithms. Detection speed is defined as the processing time of the algorithm for the input image. To evaluate the comprehensive performance of the model across diverse categories, the mean average precision (mAP) metric is utilized. It represents the average of the AP for each category, where AP is determined by analyzing the area under the precision–recall curve, which is obtained by ordering all detection results according to their confidence scores.

### 4.3. Experimental Analysis

To validate the feasibility of our proposed algorithm, we tested it on 734 test images of crayfish, and Table 3 shows the results. The precision and recall of our algorithm are 95.6% and 98.1%, respectively, with a mAP of 98.8%, a detection speed of 70.9 f/s, and a parameter size of 13.9 MB. Our algorithm possesses an accuracy of 95.6% and a recall rate of 98.1%, with an mAP of 98.8% and a parameter size measuring 13.9 MB, achieving a detection speed of 70.9 f/s.

**Table 3.** The results of the proposed algorithm.

| | Precision (%) | Recall (%) | mAP (%) | Detection (f/s) Speed | Parameters/MB |
|---|---|---|---|---|---|
| Test results | 95.6 | 98.1 | 98.8 | 70.9 | 13.9 |

Figure 10 illustrates the comparison of loss between our algorithm and YOLOv5 on the validation set. Based on the experimental findings, it was found that YOLOv5 had a lower loss function than our proposed model before epoch 169. However, after epoch 169, the loss function of our algorithm continued to decrease while that of YOLOv5 remained nearly constant. This indicates that our proposed algorithm exhibited better convergence performance, generalization ability, and model representation capacity. Based on the experimental results of this study, we successfully developed a high-performance crayfish sorting model that is highly accurate and has a fast detection speed. This model demonstrated excellent performance in the task of crayfish detection, accurately identifying the size and maturity of crayfish in the input images.

### 4.3.1. Comparative Analysis of Speed and Accuracy of Different Variants of YOLOv5

Different variants of YOLOv5 (YOLOv5s, YOLOv5m, YOLOv5l, YOLOv5n) represent models of different sizes and complexities. These variants offer a trade-off between speed and accuracy to accommodate varying computational capabilities and real-time requirements. In consideration of the quantity and distribution of the crayfish dataset, the selection process involved choosing the YOLOv5s model as the basis for our study, which was subsequently improved. To assess the feasibility of our chosen approach to improve the YOLOv5s model, we conducted comparative experiments among different variants of YOLOv5, as detailed in Table 4, where the optimal values is shown in bold.

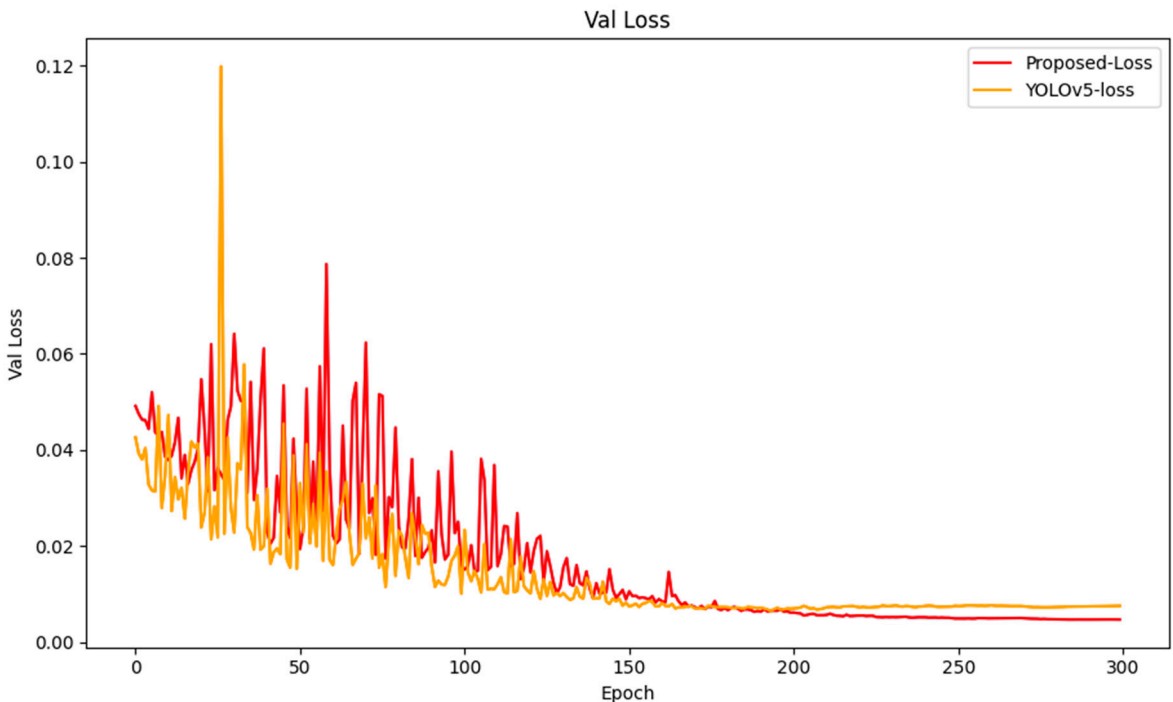

**Figure 10.** Comparison of loss between the improved model in this paper and the YOLOv5 model.

Based on the experimental results, it was found that the YOLOv5n model possesses the fewest parameters and FLOPs compared to other variants of YOLOv5. It demonstrates the highest processing speed for crayfish among the different model variations of YOLOv5. However, its precision, recall, and mean average precision (mAP) are lower compared to other algorithm groups. On the other hand, YOLOv5m and YOLOv5l demonstrate slightly higher precision, recall, and mAP. However, due to their significantly larger parameter count compared to YOLOv5n and our proposed model, their processing speed is considerably slower. Thus, we believe they are not suitable for the crayfish dataset. After comprehensive evaluation, we conclude that making improvements based on the YOLOv5s model better meets the processing requirements for crayfish.

**Table 4.** Comparative analysis of speed and accuracy of different variants of YOLOv5.

| Model | Precision (%) | Recall (%) | mAP (%) | The Processing Time per Image/ms | Parameters /10$^6$ | FLOPs /10$^9$ |
|---|---|---|---|---|---|---|
| YOLOv5n | 92.9 | 92.2 | 96.9 | **8.8** | **1.769989** | **4.2** |
| YOLOv5m | 94.9 | 95.8 | 97.4 | 17.9 | 20.899605 | 48.3 |
| YOLOv5l | 95.5 | 95.9 | 97.9 | 23.6 | 46.175989 | 108.3 |
| **Proposed** | **95.6** | **98.1** | **98.8** | 14.1 | 6.930653 | 15.7 |

### 4.3.2. Comparative Analysis of Speed and Accuracy of Different Object Detection Algorithms

Among the currently popular object detection algorithms, SSD stands out for its high speed, although it may lack precision in detection. Faster R-CNN demonstrates strong versatility but compromises accuracy due to the use of original RoI pooling with double rounding. On the other hand, Centernet achieves faster speeds by employing fewer anchors, but this reduction in anchor quantity can lead to a decrease in accuracy. The YOLO series, on the other hand, has gained widespread acclaim for its high accuracy and fast detection speed. In order to further demonstrate the superiority of our proposed algorithm in terms of crayfish recognition, we conducted qualitative and quantitative comparisons between our algorithm and eight other commonly used object detection algorithms, encompassing

SSD, Faster R-CNN, Centernet, Yolov4, Yolov5-6.0, YOLOv5-7.0, YOLOx, and YOLOv8. Consistent datasets and experimental platforms were utilized for the training and validation of all algorithms.

Figure 11 illustrates that qualitative analysis was performed on nine different object detection algorithms, encompassing SSD, Faster R-CNN, Centernet, Yolov4, Yolov5-6.0, YOLOv5-7.0, YOLOx, YOLOv8, and ours. Eight random images were selected, corresponding to the categories of green_small, green_medium, green_large, green_extralarge, red_small, red_medium, red_large, and red_extralarge. The confidence threshold of 0.5 and an IoU threshold of 0.45 were set. Figure 11 reveals the generation of false positives by the SSD algorithm, selecting a box for the blank space of the red_small category. Fast R-CNN exhibited poor performance, producing two predicted boxes for the red_small category, one of which was misclassified as green_small, resulting in low accuracy. The YOLOv4 algorithm demonstrated weak detection ability for small targets, missing the prediction box for the green_medium category. The Centernet algorithm showed poor robustness, missing the prediction box for the green_extralarge category. The YOLOv5 series, YOLOx, as well as YOLOv8, have demonstrated stable and accurate performance in predicting and testing images. This further validates the superior accuracy and robustness exhibited by the YOLO series, enabling it to reliably and accurately output results for crayfish of varying sizes and maturity levels.

As shown in Table 5, a quantitative comparison was made between these nine object detection algorithms in terms of their precision, recall, mAP, model size, and processing time per image. The algorithm proposed in this paper has the fastest detection speed among all algorithms, except for SSD. SSD was 6% faster than the algorithm presented in this paper, with a processing time per image that was 0.9 ms faster, but the mAP results demonstrated a notable 12% superiority of the proposed algorithm over SSD, and the detection speed also met the real-time requirements for crayfish detection. Therefore, while ensuring detection speed, the algorithm presented in this paper achieved the highest mAP for crayfish detection. YOLOv5-7.0 and YOLOv8 are currently considered to be highly outstanding object detection algorithms; in comparison to the algorithm proposed in this paper, they exhibit an improvement in accuracy by 0.3% and 0.9%, respectively. However, in terms of recall, mAP, model size, and speed, their performance falls short of the algorithm proposed in this study. Compared with the eight different object detection algorithms (SSD, Faster R-CNN, Centernet, YOLOv4,, YOLOv5-6.0, YOLOv5-7.0, YOLOx, and YOLOv8), the mAP of the presented algorithm was higher by 12.4%, 35.5%, 37%, 61.2%, 1.7%, 0.6%, 1.5%, and 0.5%.

This demonstrates the outstanding capabilities of the algorithm proposed for crayfish detection in terms of both speed and accuracy. In contrast to other object detection algorithms, the algorithm proposed in this study exhibited outstanding performance in both detection accuracy and robustness. This means that the proposed algorithm accomplishes the rapid and accurate detection of crayfish for different sizes and maturities, further improving the accuracy and efficiency of crayfish detection.

**Table 5.** Comparison between the proposed algorithm and other object detection algorithms.

| Model | Precision (%) | Recall (%) | mAP (%) | Model Size (MB) | The Processing Time per Image/ms |
|---|---|---|---|---|---|
| SSD | 84.5 | 89.2 | 88.0 | 94.1 | **13.2** |
| Faster R-CNN | 57.0 | 82.2 | 72.9 | 108 | 81.0 |
| Centernet | 82.5 | 58.0 | 72.1 | 124 | 14.5 |
| YOLOv4 | 66.8 | 62.5 | 61.3 | 244 | 32.1 |
| YOLOv5-6.0 | 94.6 | 94.2 | 97.1 | 14.0 | 15.9 |
| YOLOv5-7.0 | 95.9 | 96.1 | 98.2 | 14.4 | 19.8 |
| YOLOx | 94.8 | 96.6 | 97.3 | 35.1 | 18.7 |
| YOLOv8 | **96.5** | 96.1 | 98.3 | 22.0 | 17.5 |
| **Proposed** | 95.6 | **98.1** | **98.8** | **13.9** | 14.1 |

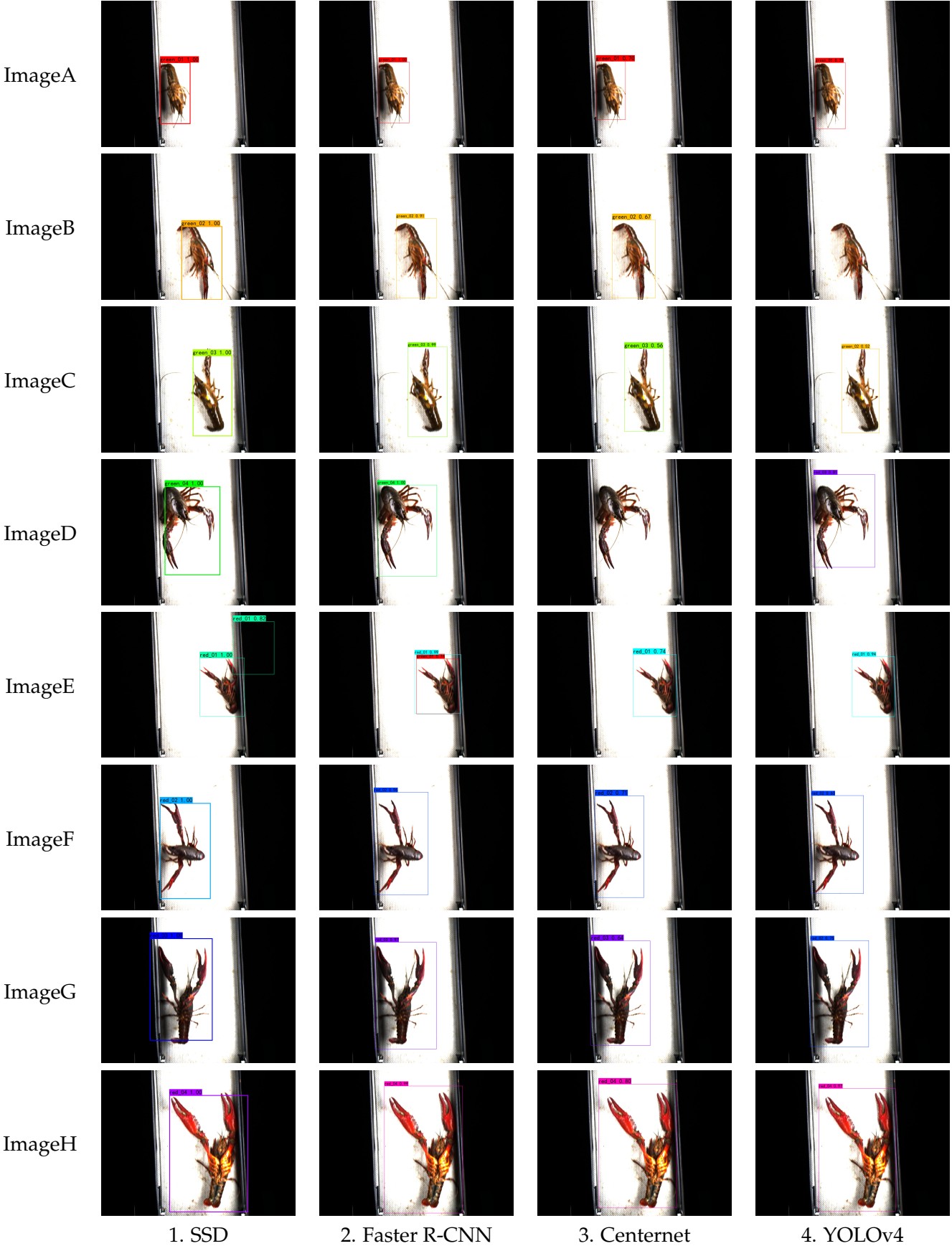

**Figure 11.** *Cont.*

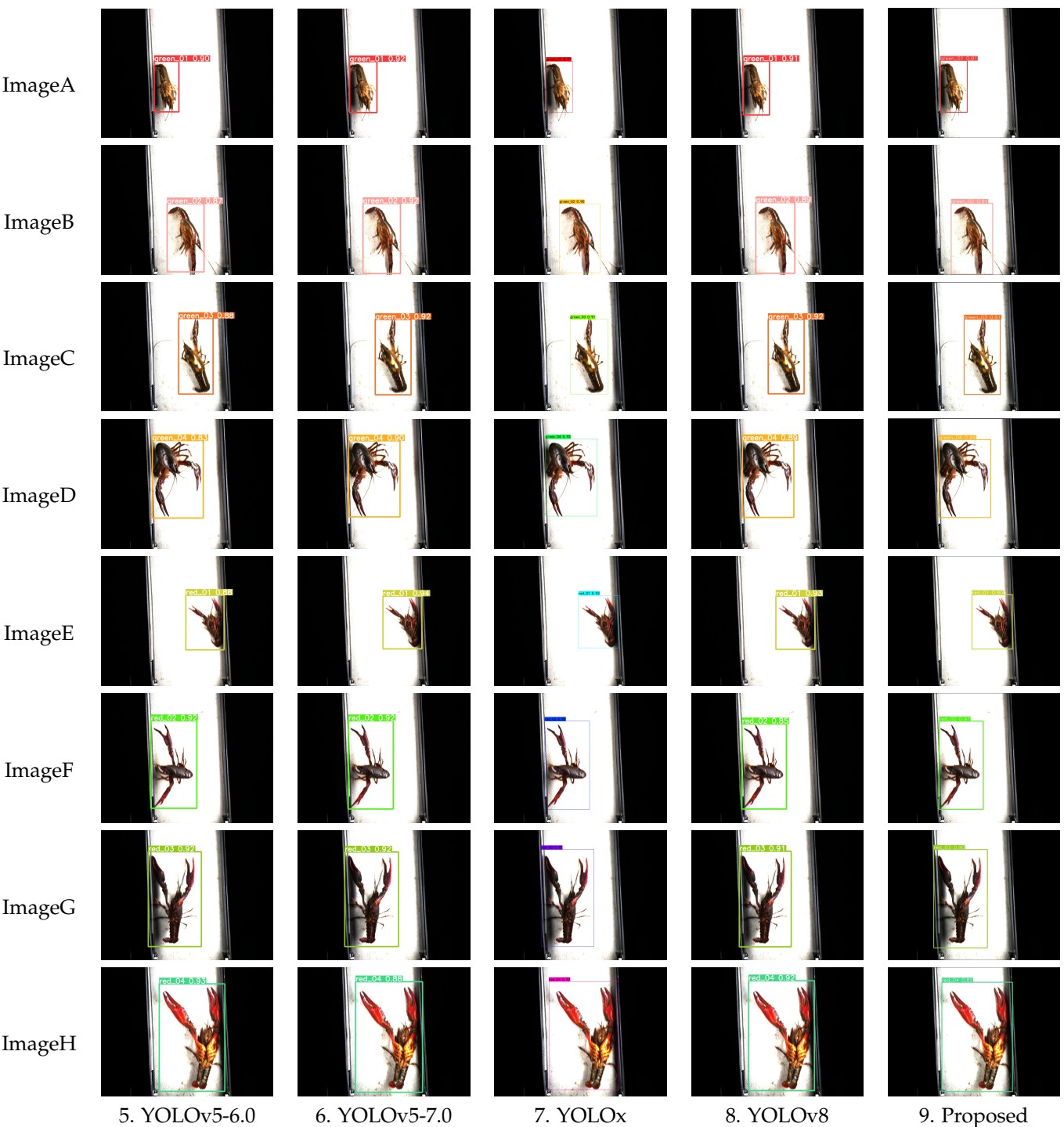

**Figure 11.** Qualitative analysis of results from various object detection algorithms.

### 4.3.3. Ablation Experiments: Comparing Speed and Analyzing Accuracy of the Algorithm

To further demonstrate the superiority of our proposed algorithm, ablation experiments were conducted using four different network configurations on the same crayfish dataset for testing. Table 6 displays the outcomes of the ablation experiments, with the most optimal results being indicated in bold font.

**Table 6.** Ablation experiments: comparing speed and analyzing accuracy of the algorithm.

| No. | Model | Precision (%) | Recall (%) | mAP (%) | The Processing Time per Image/ms | Parameters /10⁶ | FLOPs /10⁹ |
|---|---|---|---|---|---|---|---|
| 1 | YOLOv5 | 94.6 | 94.2 | 97.1 | 15.9 | 7.041205 | 16.0 |
| 2 | YOLOv5 + BoTNet | 95.5 | 96.8 | 98.5 | 17.6 | 6.714037 | 15.5 |
| 3 | YOLOv5 + CA | 95.3 | 97.1 | 98.3 | **13.1** | **5.876165** | **14.9** |
| 4 | **Proposed** | **95.6** | **98.1** | **98.8** | 14.1 | 6.930653 | 15.7 |

Since the final C3 module in the YOLOv5 model Backbone is substituted in our algorithm, with the goal of guaranteeing the accuracy of the ablation experiments, the introduced modules in the experimental groups 2, 3, and 4 were all replaced with the last C3 module in the Backbone. The parameters and FLOPs in the three experimental groups of the ablation experiment were smaller than those in the YOLOv5 model, thus proving that our algorithm has a lightweight effect. Table 6 shows that the precision, recall, and mAP of experimental groups 2 and 3 are higher than those of experimental group 1. Experimental group 2 took 2.1 ms longer to process a single image than experimental group 1, indicating that BoTNet can weaken useless features effectively and improve network performance. Group 3 had a processing time 2.8 ms shorter than group 1, demonstrating that replacing the last C3 module with a CA module could reduce network parameters and computational complexity while maintaining accuracy. Our algorithm outperforms other groups in terms of precision, recall, and mAP, with precision, recall, and mAP values surpassing those of YOLOv5 by 1%, 4%, and 1.8%, respectively, and the parameters and FLOPs are higher than those of YOLOv5. In summary, through the conducted ablation experiments, our algorithm not only achieves a significant improvement in detection accuracy but also ensures efficient detection speed.

### 4.3.4. TensorRT Deployment

To deploy the YOLOv5 model with TensorRT acceleration, first we needed to generate the .wts file from the .pt file trained with YOLOv5. Then, we compiled the C code in Visual Studio to generate the .dll file and the YOLOv5.exe file, which was used to generate the .engine file. Finally, we packaged the .engine file and .dll file to successfully deploy it using Visual Studio. Common deployment tools such as Opencv Dnn, Openvino, and Onnxruntime were also used to validate the superiority of TensorRT deployment.

Common deployment tools for object detection models include Opencv Dnn, Openvino, and Onnxruntime. To validate the excellence of TensorRT deployment, the improved YOLOv5 model was deployed on Visual Studio using these tools. Opencv Dnn and Onnxruntime were run on a GPU, while Openvino was run on a CPU. Table 7 shows the processing time for a single image.

**Table 7.** Different performance of the improved YOLOv5 model on various deployment tools.

| Tool | The Processing Time per Image/ms |
|---|---|
| Opencv Dnn-GPU | 322 |
| Openvino-CPU | 83 |
| Onnxruntime-GPU | 343 |
| TensorRT-GPU | 2~3 |

To validate the negligible impact of TensorRT deployment on accuracy, the study conducted tests using the eight test images from the previous object detection comparison experiments, as shown in Figure 12. The result revealed that the accuracy of the algorithm was not affected by TensorRT acceleration. It remained capable of accurately detecting and classifying crayfish, with an average processing speed of 3 ms per image, and a fastest speed of 2 ms per image. Therefore, the application of TensorRT for model deployment results in a marked decrease in computational complexity and greatly improves its inference speed and

performance while ensuring computational accuracy. This approach meets the real-time detection requirements for crayfish.

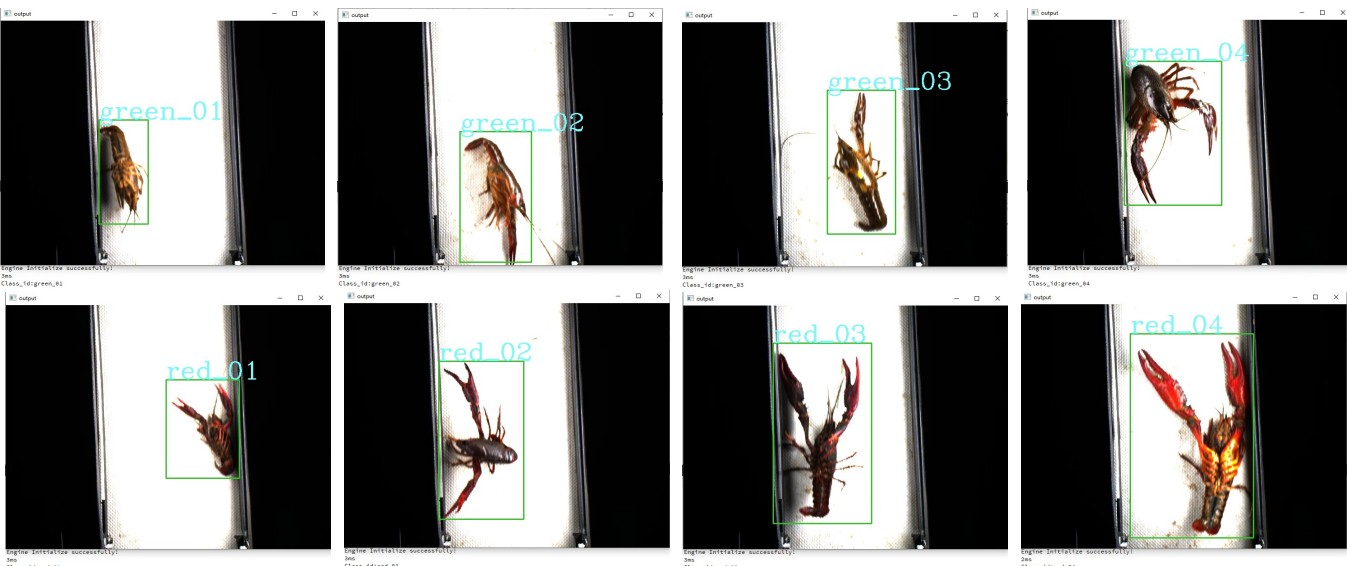

**Figure 12.** Test images for TensorRT deployment.

### *4.4. Discussion*

The proposed algorithm in this paper not only aims for accuracy but also for speed. This study is focused on improving the accuracy and speed of object detection algorithms. To achieve this goal, we compared the loss function of our improved algorithm with that of YOLOv5's, as well as compared our algorithm with various object detection algorithms in terms of both qualitative and quantitative aspects. Moreover, we conducted ablation experiments on the YOLOv5 network to compare with our proposed algorithm and validate its high precision and robustness. It is evident from the experiments that the algorithm exhibits a high precision and robustness in detecting crayfish with different sizes and maturity levels. In terms of speed, we converted the .pt file generated by the improved algorithm to a .wts file, compiled it with C language to generate a YOLOv5.exe, and used it to generate an .engine file. This file, together with the .dll file, was encapsulated to deploy with TensorRT. This approach significantly outperforms other deployment tools in terms of speed, while ensuring detection accuracy, thus achieving the high accuracy and real-time detection demands for small-sized crayfish.

### 5. Conclusions

Traditional methods for crayfish classification typically rely on manual sorting, in which workers judge the morphology of each individual crayfish and assign it to the corresponding species category. However, due to the influence of environmental factors and individual differences, this classification method is subject to certain uncertainties and subjectivity. Moreover, manual classification requires a significant amount of time and labor, is inefficient, and can easily damage live crayfish, affecting their marketability. Therefore, to achieve higher accuracy and efficiency in crayfish classification, we introduce a rapid selection method for crayfish maturity and size based on an improved YOLOv5 model. This improvement primarily involves introducing Bottleneck Transformers and Coordinate Attention to optimize the detection accuracy and robustness of the model within YOLOv5. Additionally, the application of TensorRT for model deployment results in a marked decrease in computational complexity and achieves faster speeds. It can be observed from the experimental results that our YOLOv5 algorithm achieves an mAP of 98.8% after improvements, with strong convergence and generalization capabilities and high accuracy compared to five different object detection algorithms (SSD, Yolov4, Faster R-CNN,

Centernet, Yolov5). Furthermore, after deployment on TensorRT, our algorithm model achieves detection speeds of 2 ms, enabling the rapid and high-accuracy, real-time detection of crayfish. Our method has a high-engineering application value, strong universality, and scalability, making it suitable for the classification of other marine organisms.

**Author Contributions:** Conceptualization, X.Y. and Y.L.; methodology, Y.L.; software, Y.L.; project administration, X.Y.; visualization, Y.L. and D.Z.; validation, D.Z., Y.L. and X.Y.; formal analysis, D.Z., Y.L. and X.Y.; investigation, X.H.; writing—original draft preparation, X.Y. and Y.L.; writing—review and editing, Y.L. and X.Y.; resources, Y.L. and X.H.; data curation, Y.L. and Z.H.; visualization, Y.L., Y.C. and Z.H.; supervision, D.Z. and X.Y. All authors have read and agreed to the published version of the manuscript.

**Funding:** This research was supported by the National Natural Science Foundation of China (No. 52075152), the Hubei Province Key R&D Program of China (No. 2022BBA0016), the Hubei Province agricultural machinery equipment reinforcement board core technology application project (HBSNYT202221).

**Data Availability Statement:** The dataset utilized in this study can be acquired from the corresponding author upon a reasonable inquiry.

**Acknowledgments:** We would like to thank Hubei University of Technology for providing us with a good experimental platform. We also appreciate the support of the National Natural Science Foundation of China (No. 52075152), the Hubei Province Key R&D Program of China (No. 2022BBA0016), and the Hubei Province agricultural machinery equipment reinforcement board core technology application project (HBSNYT202221). Co-first author YuXiang Liu would like to express his gratitude to his teachers and friends for their careful guidance, as well as to his family for their unwavering support, encouragement, and never-give-up attitude.

**Conflicts of Interest:** The authors declare no conflict of interest.

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
