# Peer review of "Rapid and Accurate Crayfish Sorting by Size and Maturity Based on Improved YOLOv5"

_applsci, doi:10.3390/app13158619_

Round 1
Reviewer 1 Report
The manuscript presents a Craysfish sorting system based on an improved YOLOv5 model. The authors compared existing object detectors and showed that their proposed model is superior. The manuscript is well-written and interesting to read.
Here are minor modifications suggested:
1. Caption for Figure 2 should not be centered. Please align to the left.
2. Please give more details on the captions of all the figures. Figures should be self-explanatory. For example, a bit of description of the original architecture of YOLOv5 in the caption of Figure 3 and the improvements in the caption of Figure 4. Please capitalize the first word in the captions.
3. The first paragraph of Section 4.3 Experimental Analysis repeats the same information twice: "we tested it on 734 test images of crayfish" ... "we conducted tests on a dataset of 734 crayfish images". Also, the numbers are mentioned twice.
4. The numeric results shown in Table 4 are also repeated in the text. There is no need to repeat the numbers, in the text give a summary.
Author Response
Dear Reviewer 1,
On behalf of my co-authors, we thank you very much for giving us a precious opportunity to revise our manuscript. We appreciate editor and reviewers very much for their positive and constructive comments and suggestions on our manuscript entitled “Rapid and Accurate Crayfish Sorting by Size and Maturity Based on Improved YOLOv5” (Manuscript ID: applsci-2439204).
We have carefully read the reviewer’s comments and made revisions using the revision mode. We have done our best to modify our manuscript according to the comments.
Reviewer #1:
- Caption for Figure 2 should not be centered. Please align to the left.
Respond: Thank you for your good suggestions. The caption for Figure 2 has been adjusted to left-align, and the other captions have also been left-aligned.(Located at line 203 in the revised manuscript in the word version)
- Please give more details on the captions of all the figures. Figures should be self-explanatory. For example, a bit of description of the original architecture of YOLOv5 in the caption of Figure 3 and the improvements in the caption of
Figure 4. Please capitalize the first word in the captions.
Respond: Thank you for your good suggestions. We have supplemented the captions of all the figures with additional details. Figure 3 now includes a description of the original architecture, and Figure 4's caption has been enhanced with an explanation of the improvements.(Located at lines 247 and 263, respectively, in the revised manuscript in the word version)
- The first paragraph of Section 4.3 Experimental Analysis repeats the same information twice: "we tested it on 734 test images of crayfish" ... "we conducted tests on a dataset of 734 crayfish images". Also, the numbers are mentioned twice.
Respond: Thank you for your good suggestions. We sincerely apologize for the error. The repetitive information in Section 4.3 of the experiment has been removed and improved. (Located at line 439 in the revised manuscript in the word version)
- The numeric results shown in Table 4 are also repeated in the text. There is no need to repeat the numbers, in the text give a summary.
Respond: The reviewer’s suggestions are very good for the paper modification. We have addressed the issue of duplicate numerical results appearing in both Table 4 and the text. The duplicated values in the text have been removed, and the corresponding text has been improved. (Located at line 533 in the revised manuscript in the word version)

Reviewer 2 Report
Authors Presented the article well.
1. There are so many other models were there, but author choosed very few. can you write among them why u have choosen very few.
2. Some related work should be improved by adding new work.
3. Methodology has to improve as per the objective and it should be shown clearly on results section. Proposed work was not defined clearly. Equations were not cited and they were not defined clearly.
4. Images were not clearly given, keep high resolution images. Figure 9 should be enlarged.
5. Results section should improve a lot. if possible try to show Table 4 and 5 in graphical representation.
6. Where as some more new references can be added. listed some
a. Devareddi RB, Srikrishna A. Review on Content-based Image Retrieval Models for Efficient Feature Extraction for Data Analysis. In2022 International Conference on Electronics and Renewable Systems (ICEARS) 2022 Mar 16 (pp. 969-980). IEEE.
b. Shankar RS, Srinivas LV, Neelima P, Mahesh G. A Framework to Enhance Object Detection Performance by using YOLO Algorithm. In2022 International Conference on Sustainable Computing and Data Communication Systems (ICSCDS) 2022 Apr 7 (pp. 1591-1600). IEEE.
Some grammar Mistakes were identified.
Author Response
Dear Reviewer 2,
On behalf of my co-authors, we thank you very much for giving us a precious opportunity to revise our manuscript. We appreciate editor and reviewers very much for their positive and constructive comments and suggestions on our manuscript entitled “Rapid and Accurate Crayfish Sorting by Size and Maturity Based on Improved YOLOv5” (Manuscript ID: applsci-2439204).
We have carefully read the reviewer’s comments and made revisions using the revision mode. We have done our best to modify our manuscript according to the comments.
Reviewer #2:
- There are so many other models were there, but author choosed very few. can you write among them why u have choosen very few..
Respond: Thank you for your good suggestions. The mentioned object detection algorithms in this paper, such as SSD, Faster R-CNN, Centernet, and the YOLO series, are all outstanding algorithms. We have provided additional details in the text regarding their advantages, disadvantages, and selection criteria. (For example, SSD stands out for its high speed, although it may lack precision in detection. Faster R-CNN demonstrates strong versatility but compromises accuracy due to the use of original RoI Pooling with double rounding. On the other hand, Centernet achieves faster speeds by employing fewer anchors, but this reduction in anchor quantity can lead to a decrease in accuracy. The YOLO series, on the other hand, has gained widespread acclaim for its high accuracy and fast detection speed). These algorithms have played significant roles in the field of computer vision, which is why I have chosen them. Furthermore, we have introduced other excellent object detection algorithms, such as YOLOX, YOLOv5-7.0 versions and YOLOv8, for comparison in the experimental evaluation. (Located at line 482 in the revised manuscript in the word version)
- Some related work should be improved by adding new work.
Respond: Thank you for your good suggestions. Following your suggestions, we have included other excellent object detection algorithms, such as YOLOX and YOLOv5-7.0 versions, for comparison in the experimental evaluation. The results demonstrate the superior speed and detection accuracy of the YOLO series, while also showcasing the superior speed and detection accuracy of the proposed algorithm in this paper. (Located at line 482 in the revised manuscript in the word version)
And I also added a set of experiments between different variants of YOLOv5 to verify the feasibility of improving based on the YOLOv5s model. (Located at line 458 in the revised manuscript in the word version)
- Methodology has to improve as per the objective and it should be shown clearly on results section. Proposed work was not defined clearly. Equations were not cited and they were not defined clearly.
Respond: Thank you for your good suggestions. Following your suggestions, we have made some modifications and adjustments to the titles in the experimental results and analysis to better reflect the purpose of my experiments and the results obtained(Sec 4.3.2, 4.3.3, 4.3.4). Additionally, we have provided clear summaries at the end of each experimental section to explicitly state the objectives of the experiments. Regarding the methodology and proposed work, we have summarized the aforementioned literature at the end of the Introduction section, emphasizing the need for an efficient and accurate sorting method for crayfish. (Located at line 115 in the revised manuscript in the word version) Furthermore, in the experimental results, we are able to demonstrate the effectiveness of our proposed algorithm in terms of both speed and accuracy through the presented data. Regarding equations, such as in 3.2.2, after listing the equations, we have provided explanations for each symbol in the subsequent text, including their corresponding components in the Coordinate Attention structure and an overall explanation of the workflow for the Coordinate Attention structure. (Located at line 330 in the revised manuscript in the word version)
- Images were not clearly given, keep high resolution images. Figure 9 should be enlarged.
Respond: The reviewer’s suggestions are very good for the paper modification. Following your suggestions, we have enlarged and reorganized Figure 9 to ensure high resolution. Additionally, we have provided detailed descriptions for the images in Section 3.3. (Located at line 395 in the revised manuscript in the word version)
- Results section should improve a lot. if possible try to show Table 4 and 5 in graphical representation.
Respond: Thank you for your good suggestions. Following your suggestions, I have incorporated other excellent object detection algorithms, such as YOLOX, YOLOv5-7.0 versions and YOLOv8, in the comparative experiments within the experimental results section.(Located at line 482 in the revised manuscript in the word version) And I also added a set of experiments between different variants of YOLOv5 to verify the feasibility of improving based on the YOLOv5s model. (Located at line 458 in the revised manuscript in the word version). By examining the results, we can demonstrate the superiority of the YOLO series in terms of both speed and detection accuracy, as well as highlight the superior speed and accuracy of the proposed algorithm in this paper. In each experimental section, we have conducted analyses based on the results obtained, emphasizing the superiority of the proposed algorithm in terms of speed and accuracy to demonstrate its feasibility. Regarding the presentation of Table 4 and Table 5, we have extensively reviewed similar studies and literature. Most of the referenced literature employs tables for comparative experiments. Additionally, we have highlighted Table 4, Table 5, Table 6 in purple, aiming to make it easily comprehensible for readers.
- Where as some more new references can be added. listed some
Respond: Thank you for your good suggestions. Following your suggestions, we have introduced new references into the paper.

Reviewer 3 Report
Rapid and Accurate Crayfish Sorting by Size and Maturity
Based on Improved YOLOv5
Xuhui Ye et al.
The authors develop a computer vision application based on texture and size for crayfish sorting. The application of YOLO is not novel in fish identification. However, the novelty is in deployment for crayfish sorting. The technical part of the study is solid, but there are some points which the authors should stress for this paper to become impactful.
- Why do you need to improve YOLOv5 algorithm for a small gain of speed and mAP? Does this small speed gain is required by an industrial process?
- Although mAP is the most common measure of CV models, some simple measure is needed to drive the accuracy point. Authors are requested to compare the number of green/red crayfish (as judged by a human) vs the number detected by YOLO network vs the number detected by YOLO improved model for all 4 classes in the val/test set. Is the difference significant to warrant improving the YOLO model for fish sorting?
- Why do authors need to classify the fish on its size? i.e. four different classes? Since the YOLO produces the bounding box info, it would be possible to sort the fish based on the bounding box size/area. Why this way of sorting is not sufficient? Moreover, in the proposed study, would the model predict an incorrect class if the camera is moved at a larger distance from the fish?
- The authors have studied the YOLOv5n model in this work and improved it to gain accuracy and inference speed. How do the higher YOLOv5 models perform in terms of speed and accuracy? e.g. YOLOv5s, YOLOv5m, YOLOv5l models with the original architecture.
- Have authors tried to train YOLOv8 models released in Jan 2023, known for better speed and accuracy? Does the proposed algorithm outperform YOLOv8 models?
Minor point:
- Please check the para below Sec 4.3. I think there are repeated lines.
In short, I recommend a major revision to show that this work significantly improves YOLOv5 model inference speed and accuracy, and the proposed modifications are noteworthy for industrial applications.
The paper is written in standard English.
Author Response
Dear Reviewer 3,
On behalf of my co-authors, we thank you very much for giving us a precious opportunity to revise our manuscript. We appreciate editor and reviewers very much for their positive and constructive comments and suggestions on our manuscript entitled “Rapid and Accurate Crayfish Sorting by Size and Maturity Based on Improved YOLOv5” (Manuscript ID: applsci-2439204).
We have carefully read the reviewer’s comments and made revisions using the revision mode. We have done our best to modify our manuscript according to the comments.
Reviewer #3:
- Why do you need to improve YOLOv5 algorithm for a small gain of speed and mAP? Does this small speed gain is required by an industrial process?
Respond: Thank you for your good suggestions. This gain is necessary. In actual production lines, the speed of transporting crayfish from the conveyor belt to the visual inspection device is relatively fast. A large number of crayfish will pass quickly. Therefore, it is necessary to quickly and accurately process images of crayfish and the faster and more accurate the better. The faster the visual inspection speed, the more time can be given for material sorting. The faster the visual inspection speed, the higher the cost savings. In actual crayfish production lines, crayfish sorting is divided into three parts: feeding crayfish, visual inspection, and material sorting. This paper mainly introduces the visual inspection part of crayfish. The feeding crayfish are input to the visual inspection device through a conveyor belt, and then classified crayfish are sorted by mechanical devices. Since crayfish are in short supply during peak season, it is very necessary to quickly and accurately sort and distribute them to various crayfish markets on actual production lines. Therefore, we explore an improved YOLOv5 algorithm to achieve speed and accuracy gains for industrial processes.
- Although mAP is the most common measure of CV models, some simple measure is needed to drive the accuracy point. Authors are requested to compare the number of green/red crayfish (as judged by a human) vs the number detected by YOLO network vs the number detected by YOLO improved model for all 4 classes in the val/test set. Is the difference significant to warrant improving the YOLO model for fish sorting?
Respond: Thank you for your good suggestions. We conducted experiments to verify the model performance using Precision, Recall, processing time per image, Model Size, Parameters, and FLOPs outside of mAP. We believe that these parameters are sufficient to measure the requirements of the model we need. Since crayfish are numerous in actual detection and are constantly flowing during transportation, the higher the accuracy, the fewer errors will occur. We also value speed on this basis. Our speed is five to ten times faster than manual detection. Therefore, improving the YOLO model is warranted.
- Why do authors need to classify the fish on its size? i.e. four different classes? Since the YOLO produces the bounding box info, it would be possible to sort the fish based on the bounding box size/area. Why this way of sorting is not sufficient? Moreover, in the proposed study, would the model predict an incorrect class if the camera is moved at a larger distance from the fish?
Respond: Thank you for your good suggestions. First, In China, there is a huge demand for the crayfish market, and during the peak season, there is often an imbalance between supply and demand. Among crayfish, green crayfish generally has tender meat and higher nutritional value compared to red crayfish, making them more favored. Additionally, considering transportation factors, green crayfish is more prone to mortality, hence red crayfish is typically preferred for transportation. Moreover, there is a price difference between green and red crayfish. Weight classification into four categories is a commonly used categorization method in the Chinese crayfish catering industry. For instance, smaller crayfish is used for making shrimp balls, while larger crayfish is usually prepared as oil-blistered crayfish or garlic-sautéed crayfish.
Then, after preliminary experiments, the area and weight of crayfish are not linearly proportional, so it is not just the area to measure the weight of shrimp. Using YOLO not only on the basis of extracting the area, but also taking into account other factors, we think this is the advantage of deep learning.
Finally, this study aims to pursue the practical application of the crayfish production line (as the production line must ensure its stability), and to control the environment with a certain height and light intensity to achieve high-precision and rapid sorting. In actual production lines, ensuring its light and height can ensure the accurate prediction of its model. The purpose of this paper is not to provide a model suitable for any scenario, but rather a stable and fast sorting method for use in production lines.
- The authors have studied the YOLOv5n model in this work and improved it to gain accuracy and inference speed. How do the higher YOLOv5 models perform in terms of speed and accuracy? e.g. YOLOv5s, YOLOv5m, YOLOv5l models with the original architecture.
Respond: The reviewer’s suggestions are very good for the paper modification. In our work, we studied based on the YOLOv5s model. We added an experiment to compare the performance of YOLOv5n, YOLOv5m, and YOLOv5l proposed in the experimental and analytical section.(Located at line 458 in the revised manuscript in the word version)
- Have authors tried to train YOLOv8 models released in Jan 2023, known for better speed and accuracy? Does the proposed algorithm outperform YOLOv8 models?
Respond: Thank you for your good suggestions. Yes, we conducted experiments on the crayfish dataset using the YOLOv8 model. (Located at line 491 in the revised manuscript in the word version). We introduced the results of YOLOv8 in the conclusion section of our manuscript, which is only slightly higher than the improved algorithm in this paper in terms of precision. However, other parameters are not as good as the algorithm proposed in this paper. Therefore, we believe that the algorithm we proposed is superior to the YOLOv8 model in overall consideration. We also believe that the YOLOv5 series has better stability and maturity in industrial applications and deployment.
- Please check the para below Sec 4.3. I think there are repeated lines.
Respond: Thank you for your good suggestions. We have made modifications to the paragraph below Section 4.3 and deleted the duplicate lines. (Located at line 439 in the revised manuscript in the word version)

Reviewer 4 Report
Dear authors,
here are the comments and suggestions related to the current version of your manuscript:
1. Description of all image processing techniques and transformations that were performed on images before they were used in the training process is required.
2. Provide a source code in Python that you used for image processing, training, and validation.
3. Provide more details about the serializable plan file generated in the compilation stage (section 3.3).
Author Response
Dear Reviewer 4,
On behalf of my co-authors, we thank you very much for giving us a precious opportunity to revise our manuscript. We appreciate editor and reviewers very much for their positive and constructive comments and suggestions on our manuscript entitled “Rapid and Accurate Crayfish Sorting by Size and Maturity Based on Improved YOLOv5” (Manuscript ID: applsci-2439204).
We have carefully read the reviewer’s comments and made revisions using the revision mode. We have done our best to modify our manuscript according to the comments.
Reviewer #4:
- Description of all image processing techniques and transformations that were performed on images before they were used in the training process is required.
Respond: Thank you for your good suggestions. Regarding the concerns you raised, I would like to outline my overall approach. At the crayfish farming facility, I utilized an industrial camera to capture images of the crayfish (as described in Section 2: Materials and Methods, located at line 203 in the revised manuscript in the word version). These images were then used to train an improved model, and the trained model was tested and validated (as explained in Sections 4.3.1,4.3.2,4.3.3, located at line 435 in the revised manuscript in the word version.) Experimental Results and Analysis). The results of the training, testing, and validation are all presented in the paper. Subsequently, the trained model was converted into a format compatible with TensorRT for inference. The converted model was then used for prediction and experimentation using TensorRT (as described in Section 4.3.4, located at line 608 in the revised manuscript in the word version). This encompasses the overall workflow of our image processing approach.
- Provide a source code in Python that you used for image processing, training, and validation.
Respond: Thank you for your good suggestions. For your request, I will Provide a source code in Python.
https://github.com/Tamakos1/Rapid-and-Accurate-Crayfish-Sorting-by-Size-and-Maturity-Based-on-Improved-YOLOv5
- Provide more details about the serializable plan file generated in the compilation stage (section 3.3).
Respond: The reviewer’s suggestions are very good for the paper modification. Following your suggestions, I have provided additional details in Section 3.3 regarding the serializable plan file generated in the compilation stage. (located at line 352 in the revised manuscript in the word version.) The description now includes comprehensive information about the purpose, functionality, and significance of the serializable plan file. (Batch input data is inferred through the plan file, Specifically, the enhanced YOLOv5 model (.pt) is first converted into a model format (.wts) using Python, facilitating efficient storage and utilization. Subsequently, the CMake command is executed with tensorrtx as input, resulting in the generation of a serializable file compatible with Visual Studio. The generated project is then built to produce an executable file (.exe). Additionally, the (.wts) model is converted into the (.engine) for-mat using C language, which can be read by TensorRT. Regarding the model deployment, a detection file is created within the generated project, encompassing model construction, initialization, inference, and output processes.)

Round 2
Reviewer 3 Report
I thank the authors for their detailed reply. The authors have answered all my queries. I recommend the paper for publication in Applied Sciences.
Reviewer 4 Report
Thank you for the additional explanation, information, and source code.